# Subnational Mortality Modelling: A Bayesian Hierarchical Model with Common Factors

**Qian Lu** [1], **Katja Hanewald** [2] **and Xiaojun Wang** [1,3,*]

1. School of Statistics, Renmin University of China, Beijing 100872, China; luqian2016@ruc.edu.cn
2. School of Risk & Actuarial Studies and Australian Research Council Centre of Excellence in Population Ageing Research (CEPAR), UNSW Sydney, Sydney, NSW 2052, Australia; k.hanewald@unsw.edu.au
3. Center for Applied Statistics, Renmin University of China, No. 59 Zhongguancun Street, Haidian District, Beijing 100872, China
* Correspondence: xiaojun_wang@ruc.edu.cn

**Abstract:** We propose a new model in a Bayesian hierarchical framework to project mortality at both national and subnational levels based on sparse or missing data. The new model, which has a country–region–province structure, uses common factors to pool information at the national level and within regions consisting of several provinces or states. We illustrate the model's use by drawing on a new database containing provincial-level mortality data for China from four censuses conducted during the period 1982–2010. The new model provides good estimates and reasonable forecasts at both the country and provincial levels. The model's forecast intervals reflect provincial- and regional-level uncertainty. Using subnational data for the period 1999–2018 from the Centers for Disease Control and Prevention (CDC), we also apply the model to the United States. We use mortality forecasts to compute and compare national and subnational life expectancies for China and the United States. The model predicts that, in 2030, China will have a similar national life expectancy at age 60 and a similar heterogeneity in subnational life expectancy as the United States.

**Keywords:** mortality modelling; Bayesian framework; subnational populations; life expectancy





## 1. Introduction

Many countries, both developed and developing, have experienced large reductions in mortality rates and significant improvements in life expectancy in recent decades. However, the decline in the mortality rate varies across countries, as well as across regions within countries. While subnational mortality rates and life expectancies often exhibit similar trends and levels, there are also large regional disparities. For example, in the United States, there are large inequalities in mortality (Montez et al. 2016); male life expectancy at birth in major cities such as San Francisco and Washington DC increased by 13.7 years during the period 1990–2015, whereas it only increased by 4.8 years for the entire country (Fenelon and Boudreaux 2019). In developing countries also, subnational mortality rates and life expectancies show substantial variations (e.g., Schmertmann and Gonzaga 2018 for Brazil, and Li et al. 2020 for China). The differences in life expectancy can pose challenges to the actuarial fairness of private annuity products and public pension schemes (Lee and Sanchez-Romero 2019). Therefore, subnational mortality projections are important both for insurance companies to improve the fairness of annuity products and for policymakers to evaluate pension fairness and design pension reforms.

However, subnational mortality modelling is often difficult due to limited data. For many developed countries, a relatively long time series of subnational mortality data are available (see, e.g., the United States Mortality Database, the Canadian Human Mortality Database, the French Human Mortality Database), but the available mortality datasets often have missing values at lower and higher ages at the subnational level. For developing countries, subnational mortality data are typically scarce or even unavailable.

This study proposes a new model in a Bayesian hierarchical framework to estimate and project subnational and national mortality rates based on sparse and missing data. The model uses common factors to pool information at the national level and within regions consisting of several provinces. Its forecast intervals reflect provincial- and regional-level uncertainty. We illustrate the use of the model based on a new mortality database containing data from four censuses conducted in Chinese provinces over the period 1982–2010. We also apply the model to state-level mortality in the United States based on subnational mortality data from the period 1999–2018, which has missing values at the subnational level. We project and compare the national and subnational life expectancy of China and the United States. The proposed model can also be applied to model subnational mortality and life expectancy in other countries, especially those with a limited number of data points. All model codes are available upon request from the authors.

A growing body of literature has been developing stochastic mortality models for multiple populations. For instance, Li and Lee (2005) apply the Lee–Carter model (Lee and Carter 1992) to a group of populations, allowing each population to have its own age pattern and level of mortality but imposing shared rates of change by age; this approach has had several extensions (e.g., Danesi et al. 2015; Dowd et al. 2011; Kleinow 2015). Other functional methods have also been used in modelling several related populations. For example, De Beer (2012) uses a relational model called the "tool for projecting age-specific rates using the linear splines (TOPALS)" to smooth, estimate, and project the mortality rates of 26 European countries. Hyndman et al. (2013) apply the product-ratio functional method to forecast male and female mortality in Switzerland, whereas Bergeron-Boucher et al. (2017) employ compositional data analysis to forecast mortality in 15 European countries.

A separate but related body of literature focuses on subnational mortality modelling. Some researchers have used multi-population models to model mortality in subnational areas (e.g., Gonzaga and Schmertmann 2016; Hyndman et al. 2013). Subnational mortality can also be computed applying simple relationships to national mortality forecasts (Giannakouris 2010; Office for National Statistics (ONS) (2016); Smith et al. 2013).

More recently, Bayesian methods have been used to model subnational mortality. In contrast to the Lee–Carter model or other stochastic two-stage models, which estimate parameters and forecast in two separate steps, the Bayesian framework has the following strengths. First, it has higher statistical efficiency, as it permits estimations and forecasts to be conducted in one step (Fung et al. 2017). Second, it can handle missing data. Based on this point, Alexander et al. (2017) develop a model under a Bayesian hierarchical framework to model incomplete subnational mortality rates in the United States and France. Third, the Bayesian framework is highly flexible and can adapt to almost all models. For example, Schmertmann and Gonzaga (2018) use the Bayesian framework to combine demographic estimation techniques and TOPALS in subnational mortality modelling and apply their model to Brazil.

There is limited research on developing stochastic mortality models for China, especially at the subnational level. Zhao (2012) proposes a modified Lee–Carter model for analysing short-base-period data and applies the model to (country-level) mortality data for China for the period 2000–2008. Using the model developed by Zhao (2012) and data for 2000–2008, Zhao et al. (2013) assess China's mortality rates at the country level and in three subnational groups (cities, towns, and counties). Huang and Browne (2017) propose a modified continuous mortality investigation (CMI) mortality projections model that borrows information from international experience and apply the model to country-level mortality data for China for the period 1997–2011. Applying this modified CMI model, Huang (2017) forecasts the sex–age-specific mortality rates in the same three subnational groups (cities, towns, and counties) considered by Zhao et al. (2013). Li et al. (2019) develop a Bayesian approach to handle missing data points and data from different sources to model China's (country-level) mortality based on data for 1981–2014. Using the provincial-level dataset from 1982 to 2010 introduced in this paper, Lu et al. (2020) extend the Cairns–Blake–Dowd model (Cairns et al. 2006) to a Bayesian framework and introduce the reporting probability

to study the effect of the underreporting of deaths on subnational mortality modelling and projections.

We make two main contributions to the literature. First, we propose a new model based on principal components under a Bayesian hierarchical framework to model and project subnational mortality in one stage. The proposed method allows for three geographical levels (province, region, and country) and shares information in regions through common regional and common country-level factors. The model estimates and forecasts the country- and provincial-level mortality rates simultaneously. Owing to the information-sharing structure, the new model copes with sparse and missing data. Second, we illustrate the model based on a new mortality database from four censuses for 30 Chinese provinces conducted during the period 1982–2010, which we compiled using online and archived resources. Our model captures the specific patterns within the sparse and irregular data for China. We also show that the model can be applied to more comprehensive subnational data: we apply the model to the United States using subnational data from the Centers for Disease Control and Prevention (CDC) from 1999 to 2018.

The proposed model provides a good fit and reasonable forecasts for China and its provinces, performing better, with lower values of the root mean square error (RMSE), than the Li–Lee model (Li and Lee 2005), which we choose as a benchmark. The sensitivity analyses show that the forecasts are relatively robust to the method for grouping provinces into regions. The model performs well with missing data both in China and the United States. Based on the mortality forecast, we compute the subnational life expectancy for China and the United States. Based on the currently available data, which do not cover the COVID-19 pandemic, the model projects that both countries have the same national life expectancy at age 60 and similar subnational heterogeneity. However, the life expectancy at age 60 for China has larger forecast intervals because of the relatively few data points.

The remainder of this paper is organised as follows. Section 2 introduces the proposed model. Section 3 describes the subnational mortality database for China and the United States. Section 4 presents and compares the results of the model based on Chinese subnational data. Section 5 computes and compares the national and subnational life expectancies of China and the United States. Section 6 provides conclusions and ideas for future research.

## 2. Method

The proposed model models the mortality rates at the country (national) and provincial (state) levels together, with $D_{x,t}^i$ denoting the number of deaths at age $x$ in province $i$ at time $t$ ($t = t_1, \ldots, t_T$) and $D_{x,t}^C$ representing the deaths at age $x$ in the country at time $t$. We assumed that the number of deaths follows a Poisson distribution, which is a common assumption in the literature (e.g., Czado et al. 2005; Alexander et al. 2017):

$$D_{x,t}^i \sim Poisson(m_{x,t}^i \cdot P_{x,t}^i), D_{x,t}^C \sim Poisson(m_{x,t}^C \cdot P_{x,t}^C) \tag{1}$$

where $m_{x,t}^i$ is the mortality rate and $P_{x,t}^i$ is the population at risk at age $x$ and time $t$ in province $i$, whereas $m_{x,t}^C$ and $P_{x,t}^C$ are the same variables at the country level, respectively. We applied the following consistency conditions to obtain the data for the entire country and ensured that the mortality rates at the country and provincial levels were consistent:

$$\sum D_{x,t}^i = D_{x,t}^C, \sum P_{x,t}^i = P_{x,t}^C \tag{2}$$

We modelled the provincial-level mortality rate $m_{x,t}^i$ and the country-level mortality rate $m_{x,t}^C$ as functions of principal components whose prior information is estimated by singular value decomposition (SVD).

In multi-population mortality modelling, common factors are usually used to maintain coherence (e.g., Kleinow 2015; Li and Lee 2005). We followed this approach and used common factors at different geographical levels based on the principal components. Alexander et al. (2017) suggest that the first three principal components allow for a

reasonably flexible fit and demographic interpretation at the same time and use the first three principal components in their study. Analysing Chinese subnational mortality curves, we found that three principal components provide good fits (and do not overfit) for the national and all available provincial mortality curves compared with two principal components and four principal components. Thus, we used the first three principal components as the provincial, regional and common factors in our model.

Most multi-population models emphasise coherence and model similar mortality trajectories together. However, in countries with subnational mortality disparities, such as China, all subnational areas need to be modelled together, despite the different mortality profiles. In that case, the differences in subnational mortality are substantial and can dominate the common patterns. To account for the different mortality characteristics across provinces (or states), we proposed using the first principal component as provincial-level factors to capture the dominant provincial mortality patterns. Given that provinces within a certain region experience similar levels of economic development and mortality, we introduced a regional common factor based on the second principal component to account for regional mortality similarities. The common country-level factors based on the third principal component determine the common patterns among all provinces and maintain coherence.

Following previous studies (e.g., Lee and Carter 1992; Zhao 2012), we modelled the mortality rates on the log scale. Thus, $m_{x,t}^i$ and $m_{x,t}^C$ were modelled with provincial-level, region-level, and common principal components:

$$\log(m_{x,t}^i) = \beta_{1,t}^i Y_{1,x}^i + \beta_{2,t}^k Y_{2,x}^k + \beta_{3,t} Y_{3,x} + \varepsilon_{x,t}^i \tag{3}$$

$$\log(m_{x,t}^C) = \beta_{1,t}^C Y_{1,x}^C + \beta_{2,t}^C Y_{2,x}^C + \beta_{3,t} Y_{3,x} + \varepsilon_{x,t}^C \tag{4}$$

where $Y_{1,x}^i$, $Y_{2,x}^k$, $Y_{3,x}$, and $Y_{p,x}^C$ ($p$ = 1, 2) are the principal components and $\beta_{1,t}^i$, $\beta_{2,t}^k$, $\beta_{3,t}$, and $\beta_{p,t}^C$ ($p$ = 1, 2) are the corresponding coefficients of each principal component at time $t$. The first factors, $\beta_{1,t}^i$ and $Y_{1,x}^i$, are provincial-level factors for the individual province $i$; the second factors, $\beta_{2,t}^k$ and $Y_{2,x}^k$, are common factors within the $k$th ($k$ = 1, ..., $K$) region; and the third factors, $\beta_{3,t}$ and $Y_{3,x}$, are common country-level factors for all provinces. By using common country-level factors, all provinces share the same information.

We estimated $Y_{1,x}^C$ and $Y_{2,x}^C$ by drawing from the following normal distribution:

$$Y_{p,x}^C \sim N(Y_{p,x}^{C,I}, \sigma_{y1}^2) \tag{5}$$

where $Y_{p,x}^{C,I}$ is the prior, which is the first principal component computed by the SVD of the mortality matrix for the entire country.

We modelled the provincial-level $Y_{1,x}^i$ by using the following prior distribution:

$$Y_{1,x}^i \sim N(Y_{1,x}^{C,I}, \sigma_{y1,i}^2) \tag{6}$$

where the prior $Y_{p,x}^{C,I}$ is the $p$th ($p$ = 1, 2, 3) principal component computed by the SVD of the mortality matrix for the entire country.

Provinces in the same region share the same $Y_{2,x}^k$, which is estimated as:

$$Y_{2,x}^k \sim N(Y_{2,x}^{C,I}, \sigma_{y2,k}^2) \tag{7}$$

The common $Y_{3,x}$ for the entire country and its provinces were drawn from the following distribution:

$$Y_{3,x} \sim N(Y_{3,x}^{C,I}, \sigma_{y3}^2) \tag{8}$$

The random walk process is commonly used to describe the dynamics of the period effects in multi-population and Bayesian mortality models, such as in Li and Lee (2005) and Czado et al. (2005). Therefore, we modelled the coefficients using the random walk

process. As initial values, we allowed the provincial level $\beta_{1,1}^i$ and the region level $\beta_{2,1}^k$ to pool information within region $k$:

$$\beta_{1,1}^i \sim N(\mu_{1,1}^{region\ k}, \sigma_{k1}^2), \beta_{2,1}^k \sim N(\mu_{2,1}^{region\ k}, \sigma_{k2}^2, \text{ when } t \text{ is in the } k\text{th region} \qquad (9)$$

where $\mu_{1,t}^{region\ k}$ is the population-weighted mean of the first coefficient in the $k$th region ($k = 1, 2, \ldots, K$) computed by the SVD.

The initial values of $\beta_{3,1}$ and $\beta_{p,1}^C$ and ($p = 1, 2$) pool information across all provinces with the following priors:

$$\beta_{3,1} \sim N(\beta_{3,1}^{C,I}, \sigma_3^2), \beta_{p,1}^C \sim N(\beta_{p,1}^{C,I}, \sigma_{Cp}^2) \qquad (10)$$

where the prior $\beta_{p,1}^{C,I}$ ($p = 1, 2, 3$) is the population-weighted mean computed by the SVD of the mortality matrix of all provinces. Owing to population weighting, provinces with larger populations have a more significant effect on $\beta_{p,1}^{C,I}$.

The subsequent $\beta_{1,t}^i$, $\beta_{1,t}^C$, $\beta_{2,t}^k$, and $\beta_{3,t}$ ($t > t_1$) are as follows:

$$\beta_{1,t_m}^i \sim N(\beta_{1,t_m-1}^i + \Delta_m \cdot d_1^i, \Delta_m \cdot \sigma_{e1,i}^2), \ \beta_{1,t_m}^C \sim N(\beta_{1,t_m-1}^C + \Delta_m \cdot d_1^C, \Delta_m \cdot \sigma_{e1,C}^2) \qquad (11)$$

$$\beta_{2,t_m}^k \sim N(\beta_{2,t_m-1}^k + \Delta_m \cdot d_2^k, \Delta_m \cdot \sigma_{e2,k}^2), \ \beta_{2,t_m}^C \sim N\left(\beta_{2,t_m-1}^C + \Delta_m \cdot d_2^C, \Delta_m \cdot \sigma_{e2,C}^2\right) \qquad (12)$$

$$\beta_{3,t_m} \sim N(\beta_{3,t_m-1} + \Delta_m \cdot d_3, \Delta_m \cdot \sigma_{e3}^2) \qquad (13)$$

where $\Delta_m$ is the time gap between $t_m$ and $t_{m-1}$, which means that the model allows for data to be collected at irregular time intervals (rather than annually); $d_1^i$ is the drift for province $i$, $d_2^k$ is the common drift shared by the provinces in the $k$th region, and $d_1^C$, $d_2^C$, and $d_3$ are the common drifts shared by the entire country.

The drifts are modelled from normal distributions:

$$d_1^i \sim N(d_1^{i,I}, \sigma_{d1}^2), d_2^k \sim N(d_2^{k,I}, \sigma_{d2}^2) \qquad (14)$$

$$d_1^C \sim N(d_1^{C,I}, \sigma_{d1}^2), \ d_2^C \sim N(d_2^{C,I}, \sigma_{d2}^2), \ d_3 \sim N(0, \sigma_{d3}^2) \qquad (15)$$

where $d_1^{i,I}$, $d_1^{C,I}$, and $d_2^{C,I}$ are the priors, computed by the mean of $(\beta_{p,t_m}^{C,I} - \beta_{p,t_m-1}^{C,I})/\Delta_m$ ($p = 1, 2$), $d_2^{k,I}$ is the prior computed by the regional mean of $(\beta_{2,t_m}^{i,I} - \beta_{2,t_m-1}^{i,I})/\Delta_m$ ($i \in k$th region), and $\beta_{2,t_m}^{i,I}$ is the prior provincial factor computed by the SVD of the provincial mortality.

The terms $\varepsilon_{x,t}^i$ and $\varepsilon_{x,t}^C$ are normally distributed random errors:

$$\varepsilon_{x,t}^i \sim N(0, \sigma_{\varepsilon,i}^2), \varepsilon_{x,t}^C \sim N(0, \sigma_{\varepsilon,C}^2) \qquad (16)$$

The variances are assigned conjugated inverse gamma (IG) priors, which are commonly used in the literature (see, e.g., Khana et al. 2018; Kogure et al. 2009; Li 2014). Thus we used $IG(1, 0.01)$ as priors for all assumed prior variances, for example, $\sigma_{y1}^2 \sim IG(1, 0.01)$.

Here, we considered the temporal and provincial uncertainty, and forecasted the future values of $\hat{\beta}_{1,t_T+n}^i$ and $\hat{\beta}_{1,t_T+n}^C$ by drawing from normal distributions:

$$\hat{\beta}_{1,t_T+n}^i \sim N(\beta_{1,t_T+n-1}^i + d_1^i, \sigma_{e1,i}^2), \ \hat{\beta}_{1,t_T+n}^C \sim N(\beta_{1,t_T+n-1}^C + d_1^C, \sigma_{e1,C}^2) \qquad (17)$$

where $n$ is the number of future years to forecast.

$\hat{\beta}_{2,t_T+n}^k$ and $\hat{\beta}_{2,t_T+n}^C$ are forecasted as:

$$\hat{\beta}_{2,t_T+n}^k \sim N(\beta_{2,t_T+n-1}^k + d_2^k, \sigma_{e2,k}^2), \hat{\beta}_{2,t_T+n}^C \sim N(\beta_{2,t_T+n-1}^C + d_2^C, \sigma_{e2,C}^2) \qquad (18)$$

$\hat{\beta}_{3,t_T+n}$ is forecasted by:

$$\hat{\beta}_{3,t_T+n} \sim N(\beta_{3,t_T+n-1} + d_3, \sigma_{e3}^2) \tag{19}$$

The future mortality rates of the provinces and the entire country are forecasted as:

$$\log(\hat{m}_{x,t_T+n}^i) = \hat{\beta}_{1,t_T+n}^i Y_{1,x}^i + \hat{\beta}_{2,t_T+n}^k Y_{2,x}^k + \hat{\beta}_{3,t_T+n} Y_{3,x} \tag{20}$$

$$\log(\hat{m}_{x,t_T+n}^C) = \hat{\beta}_{1,t_T+n}^C Y_{1,x}^C + \hat{\beta}_{2,t_T+n}^C Y_{2,x}^C + \hat{\beta}_{3,t_T+n} Y_{3,x} \tag{21}$$

## 3. Data

### 3.1. Subnational Mortality Data for China

We used a new database that included age- and gender-specific data on population and death by province over the period 1982–2010. China has conducted six censuses to date (1953, 1964, 1982, 1990, 2000, and 2010). Those from 1982 onward were carried out by the National Bureau of Statistics (NBS) and included population and deaths by age and gender for every province in China.[1] We obtained the data from the 1982 and 1990 censuses from hard copies in the NBS archives, and the data from the 2000 and 2010 censuses from the NBS website (NBS 2002, 2012). In addition to the censuses, China conducted four 1% sample surveys in 1987, 1995, 2005, and 2015. The 2005 and 2015 surveys contain the populations and deaths for China and its provinces. We obtained these data from the 1% Sample Survey Materials in China (NBS 2006, 2016). As the sample ratios are only around 1% for the entire country, the sample sizes at the provincial level are relatively small and the data are more volatile than the census data.

Underreporting is a concern with respect to mortality data in developing countries. China's NBS reports small underreporting ratios for population and death rates, with a maximum underreporting ratio of 1.81% for the population in 2000. Academic papers have concluded that deaths are underreported, especially for the 1990–2010 censuses; unregistered infant mortality is considered to be the main source of underreporting (e.g., Sun et al. 1993; Wang 2003; Wang and Ge 2013). The 1982 census was conducted under the guidance of the United Nations and is widely considered to be accurate and reliable (e.g., Zhai 1989; Sun et al. 1993; Li 1994). Lu et al. (2020) introduce reporting probability to study the effect of the underreporting of deaths on subnational mortality modelling and projections in China.[2]

Previous studies have shown that the data are of reasonable quality for adult ages (e.g., Banister and Hill 2004; Coale 1984; Coale and Banister 1994). Coale and Li (1991) find that the Han Chinese, the largest ethnic group in China, remember their birth year accurately because they use the lunar calendar, which assigns an animal symbol to each year in a 12-year cycle. Based on this conclusion, Zeng and Vaupel (2003) confirm that the data for the Han Chinese majority are accurate using Whipple's index and other measures. Hence, the census data have been commonly used in previous studies on mortality modelling in China (e.g., Huang and Browne 2017; Li et al. 2019; Zhao et al. 2013).

We used provincial data which were found accurate by Coale and Li (1991). We used the census data for males from 1982 onwards for the provinces to estimate our model ($T = 4$). We used the sample survey data for 2005 and 2015 as a reference for the in-sample and out-of-sample forecasts.[3]

The data in the census and sample survey materials are available in one- and five-year age groups with different maximum ages.[4] We gathered the data in five-year age groups (0, 1–4, 5–9, 10–14, . . . , 85–89, 90+) which are consistently available.

As the census enumeration is the year-end population surviving from the risk of death in the year prior to the census (Cai 2005) and the central age-specific mortality rate is

calculated based on the mid-year population, we approximated the mid-year population and accounted for deaths reported during the 12 months prior to the census as:

$$_nP_x^m = {}_nP_x + \frac{{}_na_x}{n} \cdot {}_nD_x \tag{22}$$

where ${}_nP_x$ and ${}_nP_x^m$ are the census population and mid-year population between age $x$ and $x + n$ in the census year, ${}_nD_x$ is the number of observed deaths between age $x$ and $x + n$ in the census year, and $\frac{{}_na_x}{n}$ is the average number of years lived in the year prior to the census for those who died in the year prior to the census. As in Cai (2005), ${}_na_x$ equals $\frac{n}{2}$ for all ages, except for the age groups 0 and 1–4. ${}_1a_0$ and ${}_4a_1$ were chosen according to the adapted Coale and Demeny formula (Coale et al. 1983; Preston et al. 2000).

In the proposed model, we introduced a regional level between China and its provinces. We divided China into four economic regions, as suggested by the Development Research Center of the State Council (DRC 2005) and the NBS (2011): eastern (Beijing, Tianjin, Hebei, Shanghai, Jiangsu, Zhejiang, Fujian, Shandong, Guangdong, and Hainan); central (Shanxi, Anhui, Jiangxi, Henan, Hubei, and Hunan); north-eastern (Liaoning, Jilin, and Heilongjiang); and western (Inner Mongolia, Guangxi, Chongqing, Sichuan, Guizhou, Yunnan, Tibet, Shaanxi, Gansu, Qinghai, Ningxia, ect.). Life expectancies are similar within these regions but vary across regions (Zhou et al. 2016). For example, the provinces in the eastern region have longer life expectancies than those in central and western China. These features mean that the regions are good representatives of mortality differences. We conducted sensitivity analyses on the alternative regional grouping assumptions in Section 4.6.2.

### 3.2. Subnational Mortality Data for the United States

We also applied our model to the subnational mortality of the United States. We obtained age-specific mortality data for every state within the United States from the Centers for Disease Control and Prevention dataset (CDC 2020). We used the available data for males from 1999 to 2018 for ages 0, 1–4, . . . , 80–84 years. We grouped the 50 states into four census regions, which are specified in the CDC dataset: north-eastern, midwestern, southern, and western. As described in Alexander et al. (2017), the subnational mortality data for the United States are also subject to missing data. For example, 20.6% of the death counts at age 0–10 years are missing for all states and calendar years. The missing data will be well handled by our proposed model.

## 4. Results for China

In this section, we analyse the estimation and forecast of the proposed model in detail based on new subnational mortality data for males in China. Four Chinese census datasets from 1982 onwards for 30 provinces for males aged 0 to 90+ years were used in the estimation. The proposed model will be applied to the subnational mortality data for the United States in Section 5.

We evaluated the performance of the proposed model, denoted as model $M_p$, by comparing it to the Li–Lee model (Li and Lee 2005). The Li–Lee model is widely used as a benchmark to evaluate the performance of multi-population models. Li and Lee (2005) model the mortality of multi populations using a common factor and an individual factor. To compare model $M_p$ with the Li–Lee model in one framework, we used the Bayesian framework to estimate and forecast the Li–Lee model, denoted as model $M_{LL}$. The Li–Lee model is designed for a group of populations with similar mortality rates. However, the provincial mortality rates in China have substantial variations. To compare model $M_p$ with model $M_{LL}$, we assumed that the provincial mortality rates could be modelled together in model $M_{LL}$. The original Li–Lee model proposes using a first-order autoregressive model (AR(1)) or random walk without drift to forecast the second period term. However, as our data only had a time series of 4 points with uneven time intervals, AR(1) generated discontinuous forecasts in 2011. Hence, we used the random walk without drift to forecast

the individual factor in model $M_{LL}$. Prior information for the parameters of model $M_{LL}$ was obtained by applying the original Li–Lee model. We show the estimation and forecast results for model $M_p$ and model $M_{LL}$ in the following subsections.

We completed the estimation and forecasts using R and JAGS through the R package rjags (Plummer 2019). We generated samples from the posterior distributions via the Markov chain Monte Carlo (MCMC) algorithm using Gibbs sampling. For each model, we generated samples with two chains and thinned the chains by sampling every 10th observation to reduce sample autocorrelation. The Gibbs sampling converged within 5000 iterations. After a burn-in of 20,000 iterations and convergence tests, we estimated the posterior distributions based on the last 20,000 recorded samples.

*4.1. Model Performance*

We evaluated the performance of models $M_p$ and $M_{LL}$ based on the *RMSE*, calculated as:

$$RMSE = \sqrt{\frac{\sum_{x=0}^{a} \sum_{t=t_1}^{t_T} \sum_{i=1}^{N} (\hat{m}_{x,t}^i - m_{x,t}^i)^2}{a \cdot N \cdot T}} \tag{23}$$

where $m_{x,t}^i$ is the observed mortality rate, $\hat{m}_{x,t}^i$ is the estimated mortality rate, $a$ is the number of age groups, $N$ is the number of provinces, and $T$ is the last year in the sample range. The *RMSE* values of the models are shown in Table 1. The proposed model $M_p$ has a lower *RMSE*, which indicates a better fit compared with model $M_{LL}$.

**Table 1.** *RMSE* of models in census years.

| Model | $M_p$ | $M_{LL}$ |
|---|---|---|
| *RMSE* | $6.39 \times 10^{-3}$ | $7.44 \times 10^{-3}$ |

*4.2. Parameters*

Figure 1 shows the estimated parameters of two representative provinces:[5] Beijing, a developed eastern province; and Gansu, a less developed western province. The white lines and grey shading show the estimated medians and the 95% posterior intervals of Beijing, respectively. The black lines show the estimated median of Gansu. The dashed lines delineate the corresponding 95% posterior interval.

Figure 1 illustrates that the proposed model allows for province-specific estimated values for $Y_p^i$ and $\beta_{p,t}^i$ ($p = 1, 2$) and assumes the same values for $\beta_{3,t}$ and $Y_3$ because of the information-sharing structure. The first principal component $Y_1^i$ reflects the overall shape of the mortality curve. Gansu has higher $Y_1^i$ than Beijing at ages 20–40, but lower $Y_1^i$ at age 80+ (which is likely due to underreported mortality data for Gansu). The effect of underreported mortality data in Gansu is discussed in Lu et al. (2020). Beijing has a higher and steeper $\beta_{1,t}^i$, indicating a faster improvement in overall mortality in the eastern provinces than in the western provinces. The second principal component $Y_2^k$ reflects the common regional difference in mortality compared to the overall mortality curve. For Beijing and Gansu, the shapes of $Y_1^k$ are similar. Beijing has a higher level of $\beta_{2,t}^k$ than Gansu, whereas Gansu has a steeper trend in $\beta_{2,t}^k$ than Beijing. The third factors, $Y_3$ and $\beta_{3,t}$, reflect the common national differences in mortality compared with the regional differences and the overall mortality curve—they are the same for all provinces.

The results for the remaining provinces can be summarized as follows: different provinces have different levels of $\beta_{1,t}^i$ and different shapes of $Y_1^i$; provinces in different regions have different levels of $\beta_{2,t}^k$ and different shapes of $Y_2^k$; and all provinces have the same $\beta_{3,t}$ and $Y_3$.

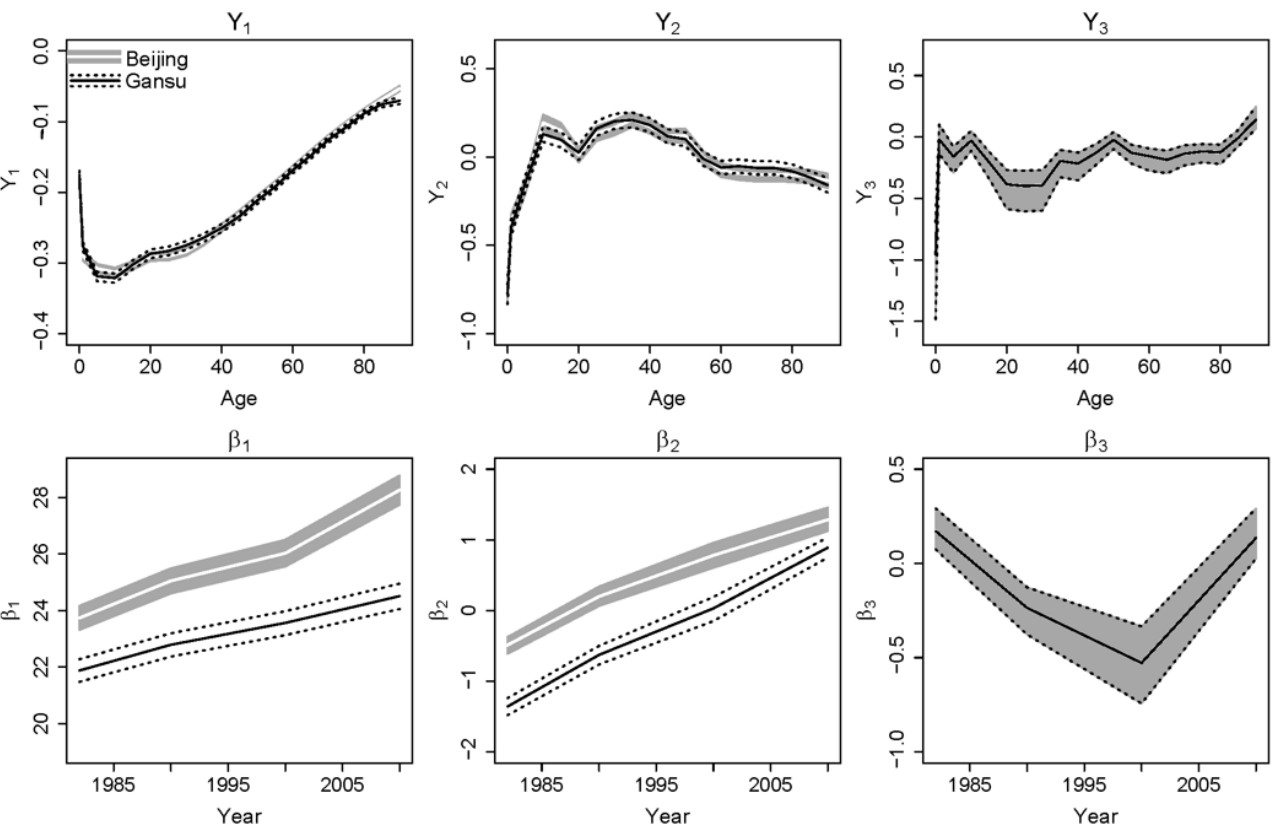

**Figure 1.** Estimated parameters (median and 95% posterior intervals) for Beijing and Gansu based on the proposed model.

### 4.3. Estimation Results

Our proposed model $M_p$ estimates national and subnational mortality simultaneously, whereas the Li–Lee model $M_{LL}$ only estimates subnational mortality. Thus, we used the population-weighted average to calculate the national mortality estimates for model $M_{LL}$. As the estimation results are similar for all four census years and provinces, we show the results for China, Beijing, and Gansu in the first and last census years.

The fan charts in Figure 2 show the estimated mortality rates from the proposed model and the Li–Lee model $M_{LL}$ for China, Beijing, and Gansu in census years 1982 and 2010. The black dots are the historical data, the grey intervals are the 95% posterior intervals of the proposed model, and the intervals between the black dashed lines are the 95% posterior intervals of model $M_{LL}$. Figure 2 shows that the proposed model reproduces the historical mortality rates of China and the provinces well and covers the historical data better than model $M_{LL}$.

We used mortality data from the 1% sample survey data for China and its provinces to assess the quality of in-sample and out-of-sample forecasts. As noted in Section 2, the 1% sample survey data are relatively volatile at the province level due to small sample sizes, making the random errors dominate the goodness of fit. Keeping regions identical to those mentioned previously, we used the average data of every region in the sample survey to reduce the data volatility and analyse the in- and out-of-sample forecasts. The regional average of the sample survey data is weighted and calculated by the total deaths divided by the total population at risk within the region: $\log\left[\left(\sum_{x=0}^{a}\sum_{i=1}^{r_k} D_{x,t^*}^i\right)\middle/\left(\sum_{x=0}^{a}\sum_{i=1}^{r_k} P_{x,t^*}^i\right)\right]$, where $r_k$ is the number of provinces within the region and $t^*$ is the year of the sample survey. The regional average estimations of model $M_p$ and model $M_{LL}$ are calculated by the population-weighted average of the estimated log-scale mortality within the region.

The larger the region, the less volatile the regional average. Therefore, we show the average of the two largest regions, the eastern and western regions, and the corresponding

sample survey data in 2005 in Figure 3. The hollow dots are the regional average of the sample survey data; and the grey intervals and the intervals between the black dashed lines are the 95% posterior intervals of the regional average of the proposed model and model $M_{LL}$, respectively. The proposed model has higher estimates of mortality rates for ages 10–40 in the western region than the Li–Lee model $M_{LL}$ and reflects mortality patterns in the western region better. We conclude that both models $M_p$ and $M_{LL}$ generate reasonable in-sample forecasts for China and its regions.

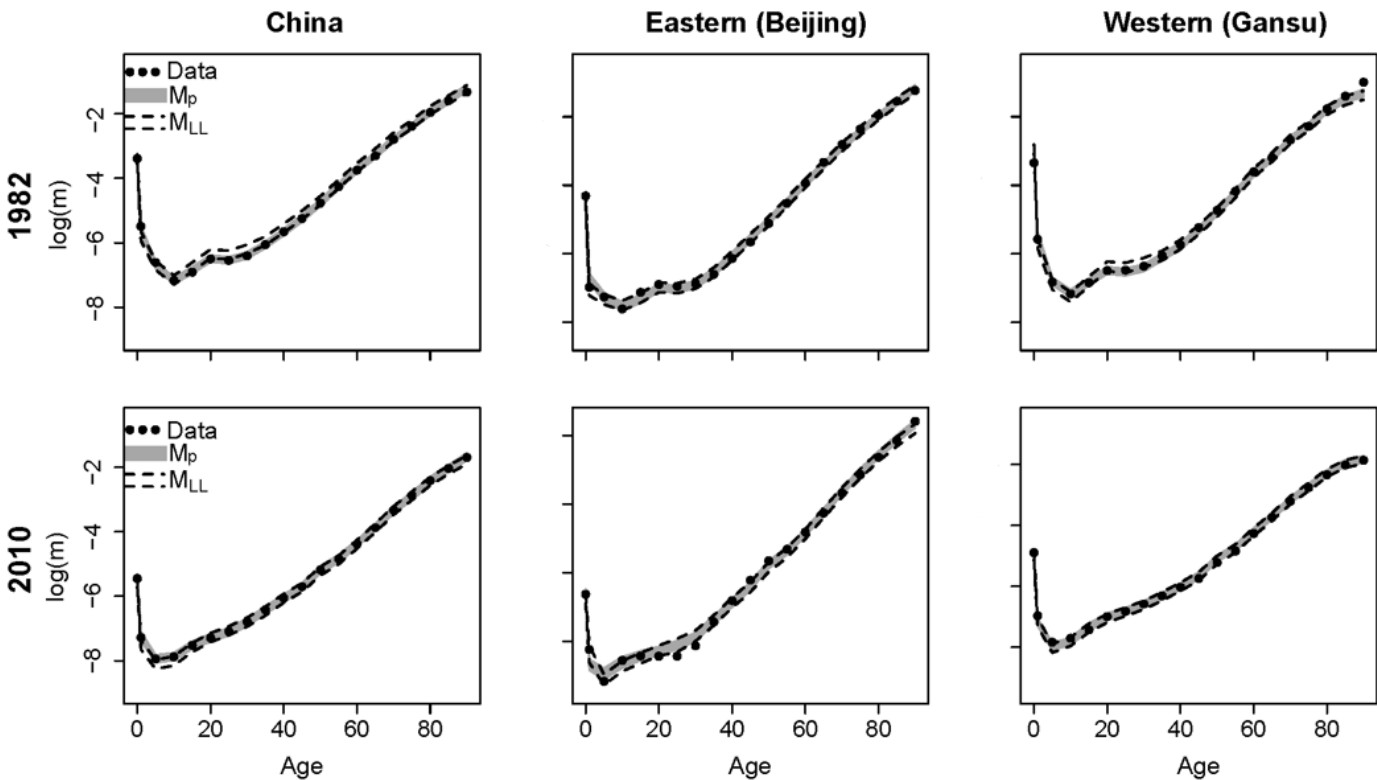

**Figure 2.** 95% posterior intervals for China and two provinces in 1982 and 2010.

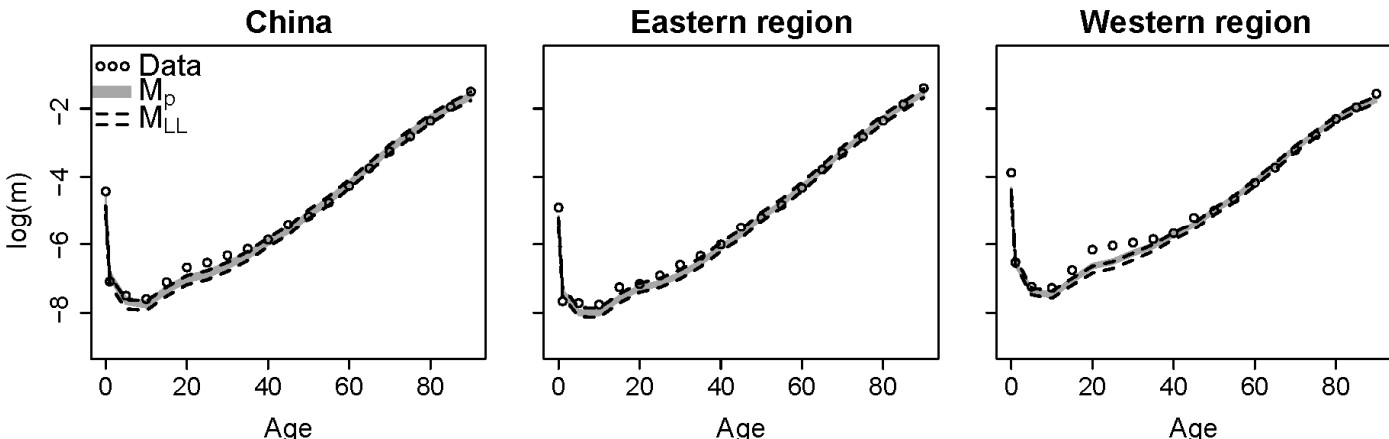

**Figure 3.** In-sample forecasts for China and two regions in 2005.

*4.4. Performance on Missing Data*

　　The proposed model handles missing data well. Although the provincial mortality data for Tibet in 1982 are missing, the models can still estimate mortality using information from the other provinces. Figure 4 shows the estimation results for Tibet in 1982. The white

lines show the estimated median mortality rates and the grey interval is the corresponding 95% posterior interval for the proposed model. The intervals are wider than for other provinces (e.g., the provinces shown in Figure 2) because of larger uncertainty due to missing data.

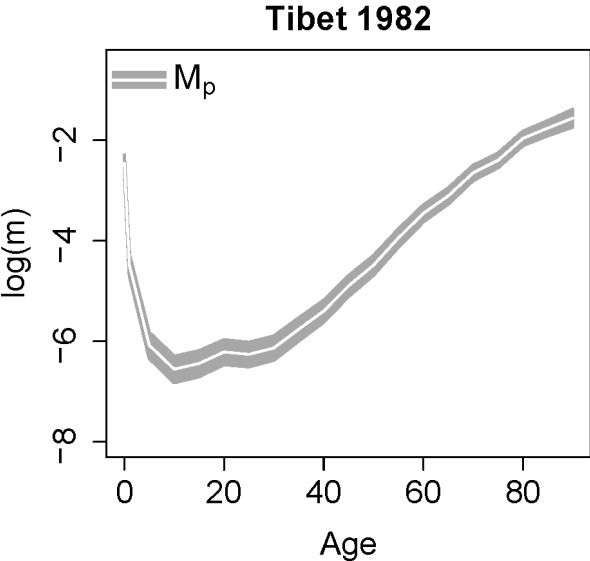

**Figure 4.** Estimated mortality rates for Tibet in 1982.

### 4.5. Model Forecast

We used the national and regional average mortality of the 2015 sample survey data to evaluate the out-of-sample forecast performance of the proposed model. Figure 5 shows the out-of-sample forecast for China and its eastern and western regions in 2015. The proposed model generates narrower forecast intervals than the Li–Lee model $M_{LL}$. Table 2 shows the RMSE values of the proposed model and model $M_{LL}$ in 2015. The proposed model has a lower RMSE for China and its regions than model $M_{LL}$.

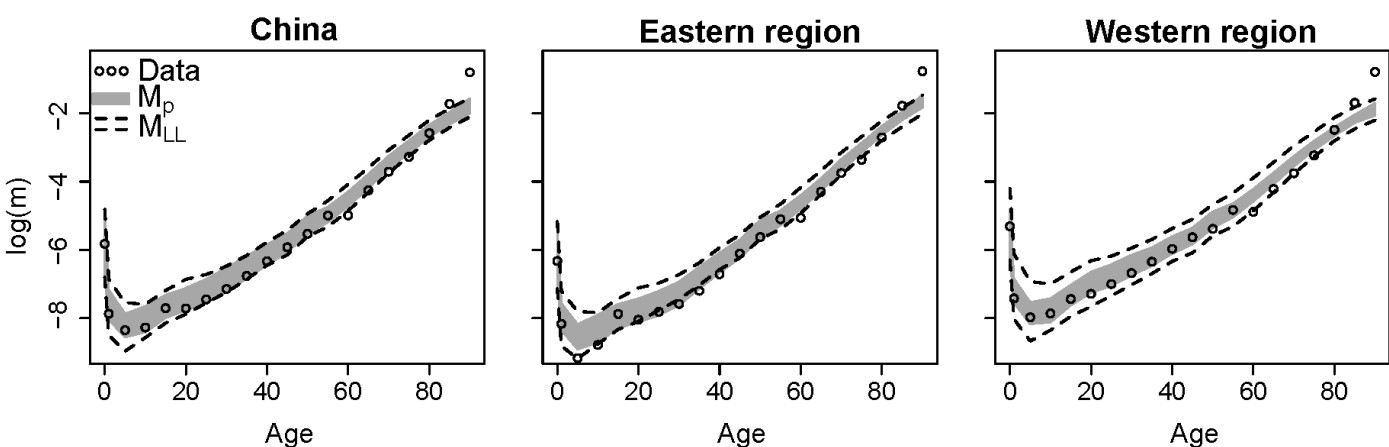

**Figure 5.** Out-of-sample forecasts for China and two regions for 2015.

**Table 2.** RMSE of proposed model $M_p$ and Li–Lee model $M_{LL}$ for forecasts to 2015.

|  | $M_p$ | $M_{LL}$ |
|---|---|---|
| China | $6.25 \times 10^{-2}$ | $6.49 \times 10^{-2}$ |
| Eastern region | $5.84 \times 10^{-2}$ | $6.27 \times 10^{-2}$ |
| Western region | $6.63 \times 10^{-2}$ | $6.70 \times 10^{-2}$ |

Figure 6 displays the estimation and forecast results for the different age groups in the four provinces representing the four regions.[6] The proposed model and the Li–Lee model $M_{LL}$ have different estimates, forecast trends, and forecast intervals at younger ages (below age 4). However, the estimates and forecast trends are similar at other ages.

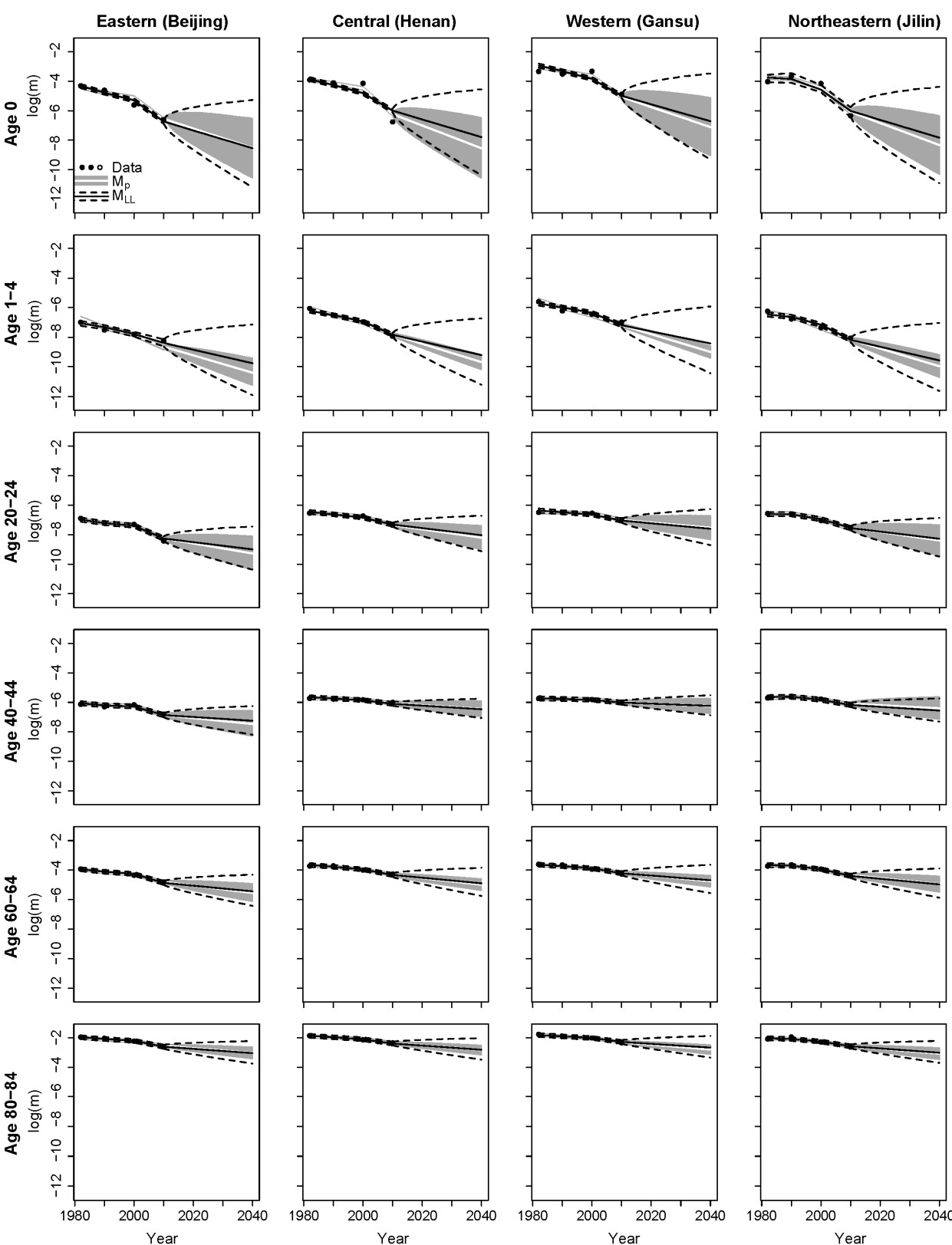

**Figure 6.** Estimates and forecasts of the proposed model $M_p$ and the Li–Lee model $M_{LL}$.

As the proposed model uses provincial and regional uncertainty in forecasts, the intervals in Figure 6 show that the provincial forecasts of the proposed model have smaller uncertainty than model $M_{LL}$, which uses the overall uncertainty of all provinces. The forecast intervals of both models show that infant mortality has the largest uncertainty, which decreases as the age increases.

The provincial forecasts of the proposed model have different levels of uncertainty for the same age groups. When comparing different regions, the eastern region (the first column) and the north-eastern region (the last column) have larger uncertainties, indicating that provincial variations are also larger in these two regions. However, the forecasts for the central region (the second column) and the western region (the third column) have lower uncertainties, indicating smaller provincial variations in these regions. Moreover, model $M_{LL}$ has larger and equal widths of intervals for different regions. From the forecast uncertainty perspective, the proposed model captures different levels of uncertainty for the different regions.

The estimates for 1982–2010 and the forecast results for 2011–2040 for the entire country are shown in Figure 7. The proposed model generates a better fit for the historical trends than model $M_{LL}$. The proposed model has steeper forecast trends than model $M_{LL}$, especially for infants and young children (aged 1–4 years). Although the proposed model has narrower forecast intervals than model $M_{LL}$, both the proposed model and model $M_{LL}$ cover the 2015 sample survey data equally well.

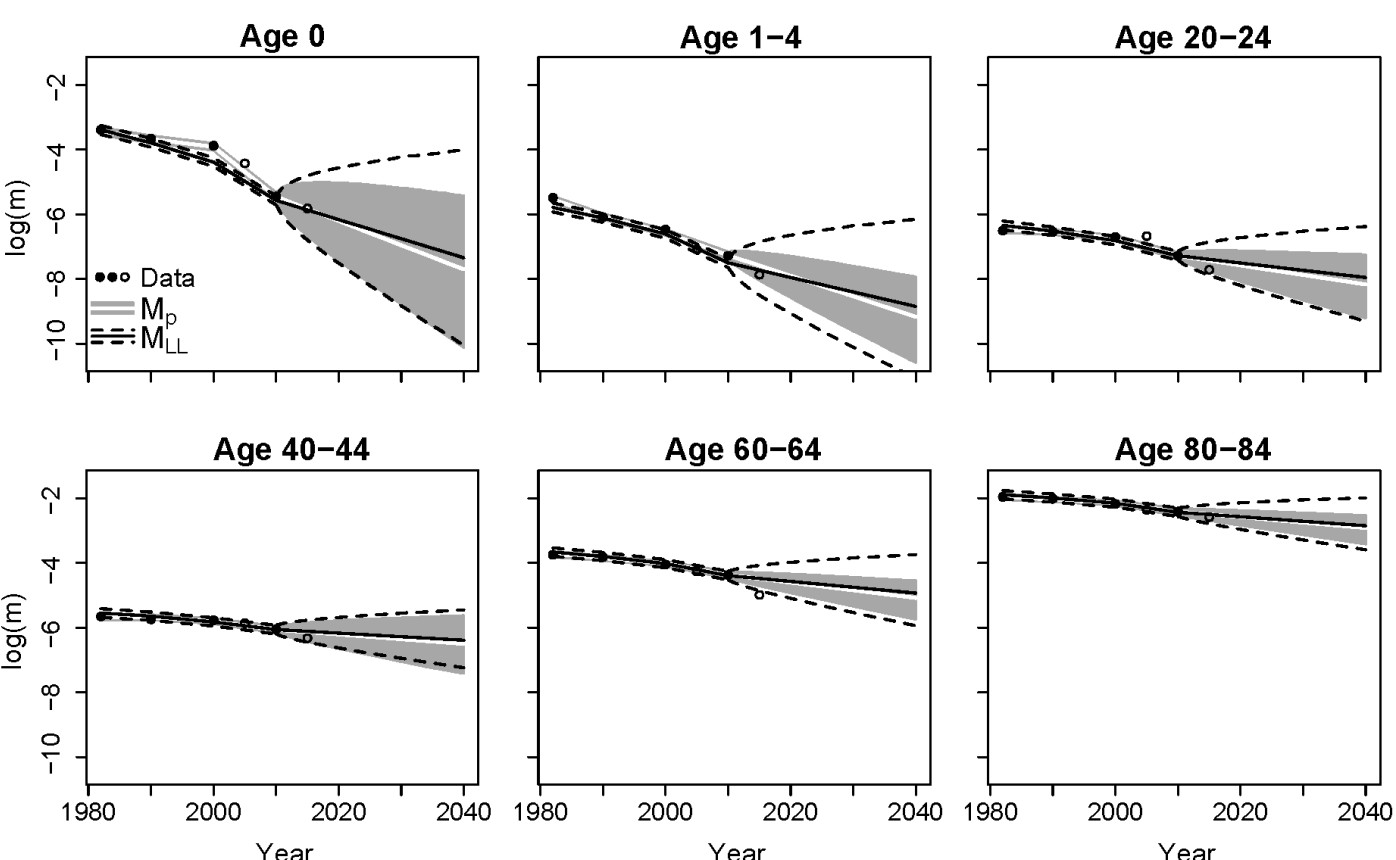

**Figure 7.** Estimates and forecasts for China.

### 4.6. Sensitivity Analysis

4.6.1. Grouping Assumption

In the main analysis, we used the official classification developed by the NBS to group the provinces into regions based on their geographical and economic characteristics. However, this regional grouping assumption may be questioned, and other grouping

assumptions may lead to different forecasts. As such, we conduct sensitivity analyses on alternative regional grouping assumptions in the following.

In the main analysis, Hebei province is included within the most developed (i.e., eastern) region in our proposed model. However, Hebei's economic development (e.g., GDP per capita; NBS 2019) and life expectancy (Zhou et al. 2016) are similar to those of less developed regions. Since Hebei lies adjacent to the central, north-eastern, and western regions, it can be included within any of these groups. In the following, we use different grouping assumptions to derive forecasts for the proposed model with a random walk process. Specifically, we compare the results based on the original grouping, where Hebei is part of the eastern region, with three alternative grouping assumptions: (1) Hebei is in the central region; (2) Hebei is in the western region; and (3) Hebei is in the north-eastern region.

Figure 8 shows the forecast results based on the original grouping and the other three grouping assumptions. The forecast results are similar for the different age groups, so we consider the results at ages 70–74 for Guangdong as representative of the eastern region. We also show the results from Heilongjiang (which is in the north-eastern region) in Figure 8 as the north-eastern region has the least number of provinces and it is interesting to observe the changes when Hebei is included.

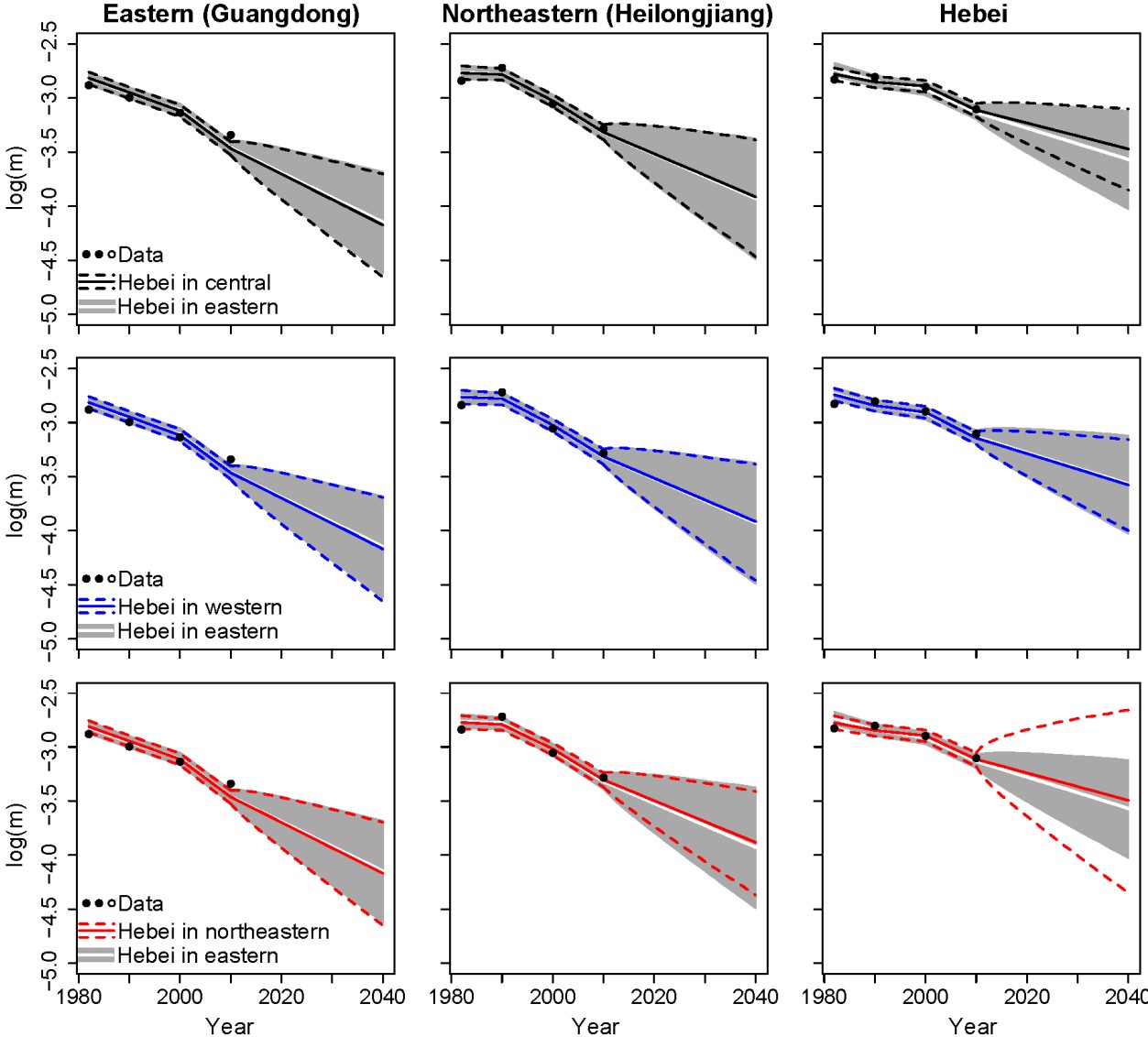

**Figure 8.** Results of different grouping assumptions in the proposed model (for ages 70–74).

The first column in Figure 8 shows that the forecast trends and intervals are basically the same when Hebei is excluded from the eastern region. The second column in Figure 8 shows that, when Hebei is included in the north-eastern region, the forecast trend of Heilongjiang is slightly less steep and the forecast interval is slightly narrower than in the original grouping. Overall, the differences are limited when the grouping assumption is changed. The third column in Figure 8 shows that the fit and forecasts for Hebei change according to which region it is included in. When Hebei is included in the central region, the slope of the forecast trend is less steep than in the original grouping, whereas the forecast interval is narrower than the original grouping assumption and other new grouping assumptions, indicating that Hebei's mortality rates are more similar to those in the central region.

### 4.6.2. Number of Principal Components

In the main analysis, the proposed model is constructed with three principal components. Here, we compare the model performance of the proposed model and an alternative model constructed using two principal components (denoted as $M_{PC2}$).

Based on the structure of the proposed model described in Section 2, the alternative model constructed with two principal components (model $M_{PC2}$) is given as follows:

$$\log(m_{x,t}^i) = \beta_{1,t}^i Y_1^i + \beta_{2,t} Y_2 + \varepsilon_{i,t}^i \tag{24}$$

$$\log(m_{x,t}^C) = \beta_{1,t}^C Y_1^C + \beta_{2,t} Y_2 + \varepsilon_{i,t}^C \tag{25}$$

where $\beta_{2,t}$ and $Y_2$ are the common factors across all provinces. The parameters are estimated and forecasted as for the main proposed model described in Section 2.

We compare the RMSE of the alternative model $M_{PC2}$, the proposed model $M_p$, and the Li–Lee model $M_{LL}$ in Table 3. The alternative model $M_{PC2}$ has larger errors in fittings and forecasts than the proposed model and model $M_{LL}$. Figure 9 shows the estimations and forecasts of $M_{PC2}$ and $M_p$. The blue solid and dashed lines are the fitting and forecast intervals of model $M_{PC2}$, respectively. The proposed model $M_p$ reproduces better historical mortality trends than the alternative model $M_{PC2}$, especially for ages 0–4. $M_{PC2}$ generates narrower forecast intervals than $M_p$.

**Table 3.** RMSE of models.

|  |  | $M_p$ | $M_{LL}$ | $M_{PC2}$ |
|---|---|---|---|---|
| Census (China) |  | $6.39 \times 10^{-3}$ | $7.44 \times 10^{-3}$ | $7.60 \times 10^{-3}$ |
|  | China | $6.25 \times 10^{-2}$ | $6.49 \times 10^{-2}$ | $6.56 \times 10^{-2}$ |
| 2015 | Eastern region | $5.84 \times 10^{-2}$ | $6.27 \times 10^{-2}$ | $6.30 \times 10^{-2}$ |
|  | Western region | $6.63 \times 10^{-2}$ | $6.70 \times 10^{-2}$ | $6.80 \times 10^{-2}$ |

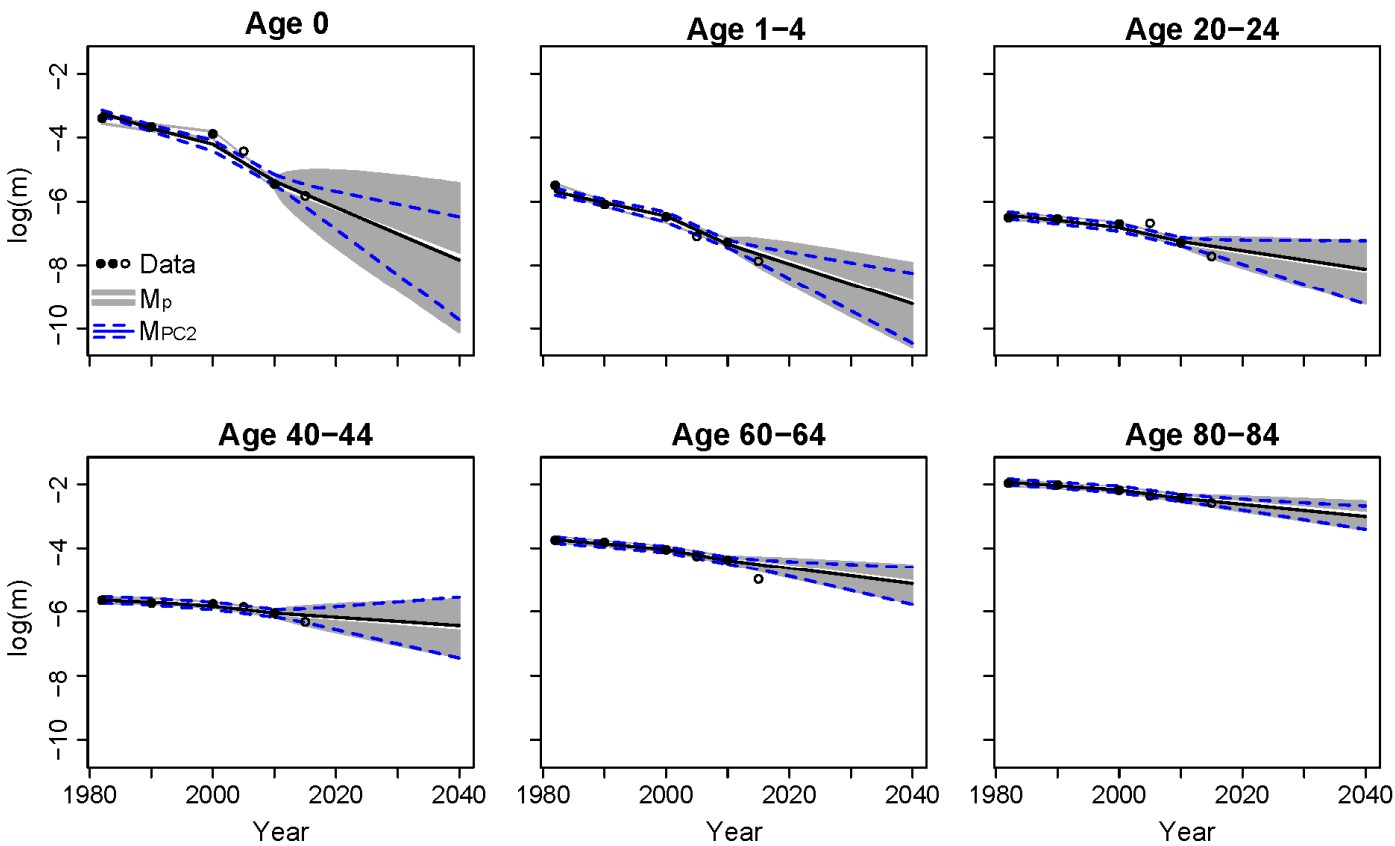

**Figure 9.** Estimates and forecasts of $M_{PC2}$ and $M_p$ for China.

## 5. Life Expectancy in China and the United States

### 5.1. Life Expectancy in China

Based on the provincial mortality forecast of model $M_p$ for China, we calculated the provincial life expectancy. Life expectancy at pension eligibility age is an important reference for pension systems. In China, the normal retirement age for males is 60. In the following, we calculate the provincial-level life expectancy at age 60 for China and evaluate the subnational heterogeneity in life expectancy.

Figure 10 shows the provincial life expectancy at age 60 in China ($LE_{60}^{i,C}$) in 2030. The areas shaded grey are the 95% intervals of $LE_{60}^{i,C}$, the darker grey short lines are the medians of $LE_{60}^{i,C}$, and the grey dashed line is the national $LE_{60}^{C}$ calculated based on the national mortality projection. The model predicts that the median $LE_{60}^{C}$ will be 24.6 years in 2030 (up from to 22.5 years in 2020).

The provincial in 2030 varies across regions. Most of the provinces in the eastern region have a higher $LE_{60}^{i,C}$ than the national $LE_{60}^{C}$, whereas most of the provinces in the western region have a lower $LE_{60}^{i,C}$ than the national $LE_{60}^{C}$. Hainan, a southern island and holiday destination, has the highest median $LE_{60}^{i,C}$ of 27.7 years. The provinces in the north-eastern region have wider intervals than others.

To summarize, we calculated the national and subnational life expectancy at retirement in 2030 for China based on subnational mortality forecasts. The median life expectancy at pension eligibility age in China ($LE_{60}^{C}$) is 24.6 years in 2030, with the median provincial life expectancy $LE_{60}^{i,C}$ ranging from 21.3 years to 27.7 years. In the following subsection, we apply our proposed model to estimate subnational life expectancies in the United States and compare them with China.

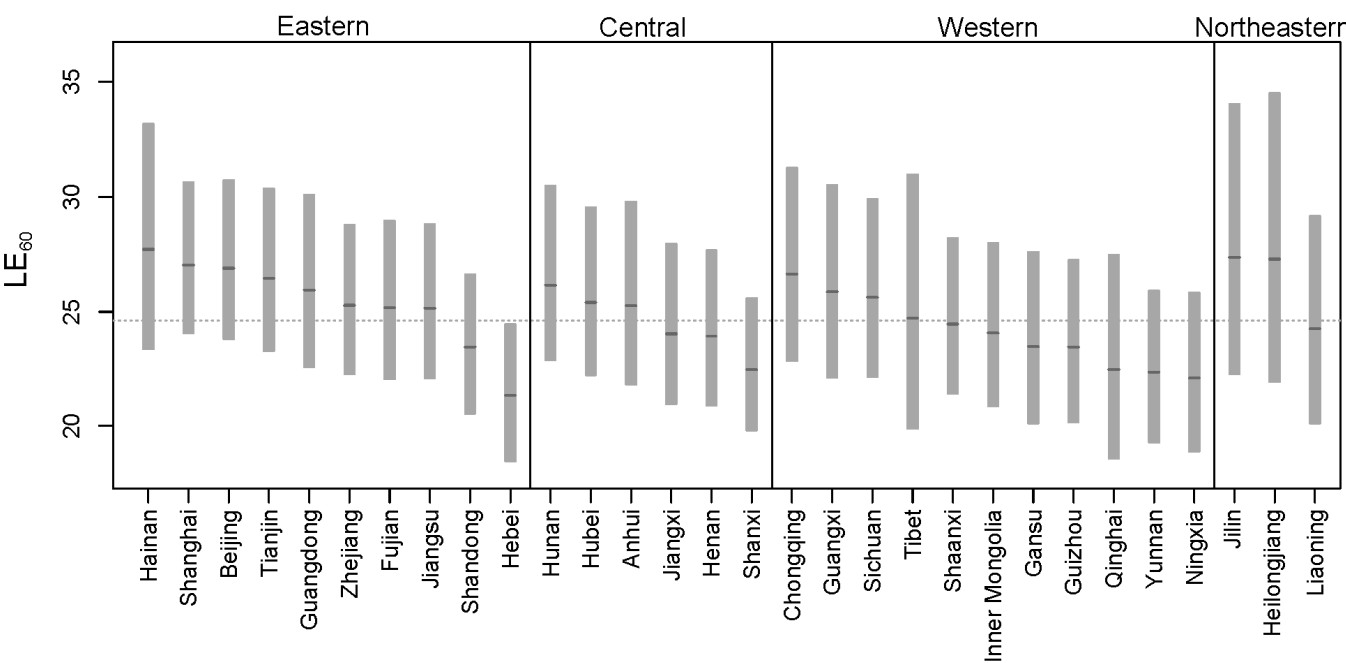

**Figure 10.** $LE_{60}^{i,C}$ of Chinese provinces in 2030.

## 5.2. Life Expectancy in the United States

We applied the proposed model described in Section 2 to subnational mortality in the United States based on five-year age group data (0 to 80–84 years old) from 1999 to 2018. After obtaining estimations and projections of subnational mortality rates in the United States, we used the Kannisto model (Kannisto 1994) to extrapolate the mortality rates to age 90+ years and calculate the national and subnational life expectancy in the United States.

Figure 11 shows the projected state-level life expectancy at age 60 for the United States ($LE_{60}^{i,US}$) in 2030. The grey shaded areas are the 95% intervals of , the darker grey short lines are the medians of $LE_{60}^{i,US}$, and the grey dashed line is the national $LE_{60}^{US}$ projected by the model. In 2030, the median $LE_{60}^{US}$ is 24.9 years (up from to 23.2 years in 2020), which is 0.3 years higher than China.

The median ranges from 21.6 to 27.5 years. Most of the states in the western region have a higher $LE_{60}^{i,US}$ than $LE_{60}^{US}$, with Hawaii having the highest $LE_{60}^{i,US}$. However, most of the states in the midwestern and southern regions have a lower $LE_{60}^{i,US}$ than $LE_{60}^{US}$.

China has the same level of national life expectancy at age 60 as the United States. Based on the model results, in 2020, the national life expectancy at age 60 is 22.5 and 23.2 years for China and the United States, respectively. In 2030, predicted by our model, the national life expectancy at age 60 will be 24.6 and 24.9 years for China and the United States, respectively. In 2030, the median subnational life expectancy at age 60 for these two countries is at the same level, with 21.3 and 27.7 years, and 21.6 and 27.5 years for China and the United States, respectively. The comparison indicates that in 2030, China will approach the same level of life expectancy at age 60 as the United States. However, the forecast intervals of subnational life expectancy at age 60 are wider in China than they are in the United States due to fewer available historical data points.

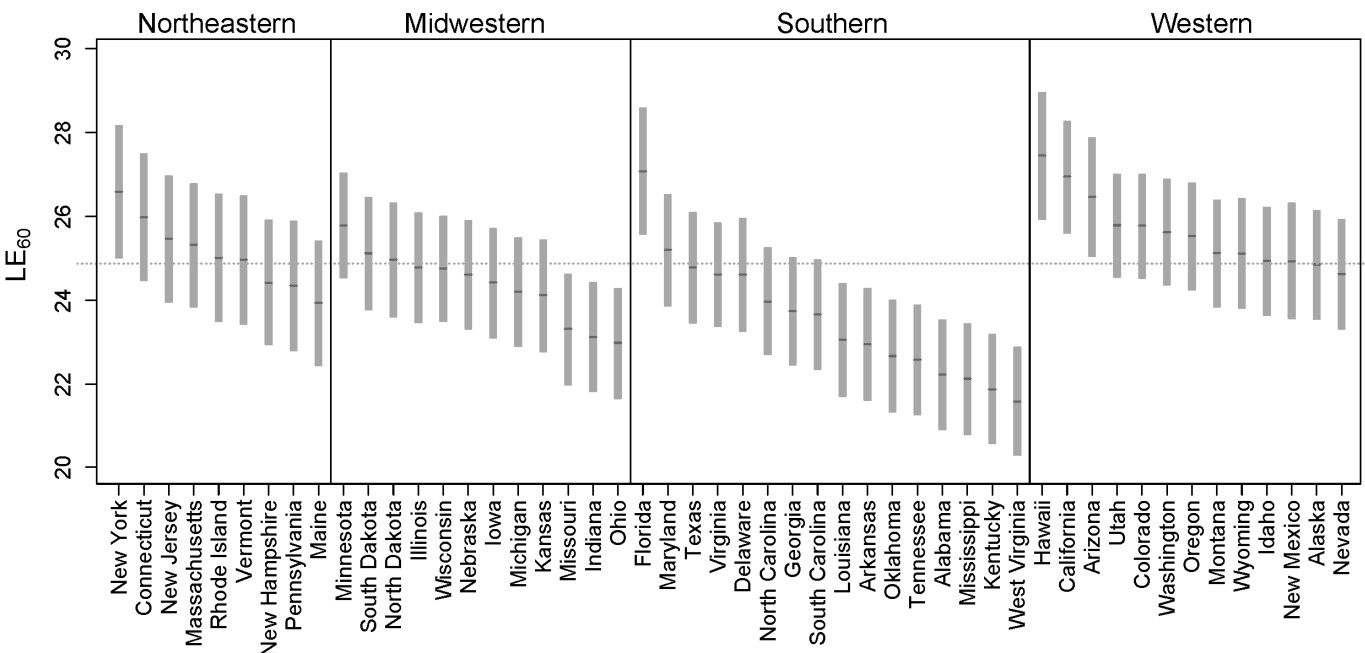

**Figure 11.** The $LE_{60}^{i,US}$ in 2030.

## 6. Conclusions

This paper describes a new model in a Bayesian hierarchical framework to estimate subnational mortality rates and forecast mortality at both subnational and national levels. We propose a Bayesian hierarchical model based on principal components and random walk processes. The model uses the three levels of country, region, and province, with the information pooling and sharing structure through provincial-level factors, common region-level factors, and common country-level factors. Our study employs a new database of provincial-level mortality data for China from 1982 to 2010 and uses US subnational mortality data from 1999 to 2018. We note that the available time series for provincial mortality data in China are very limited and that both China and the United States have missing data for some provinces/states.

We evaluated the performance of our proposed model in detail based on new subnational mortality data for China. The evaluation results show that the proposed model copes with missing data and provides a good fit for the census and sample survey years, and reasonable forecasts at both the provincial and country levels. The forecast intervals are of equal width for all provinces within any one age group because of the model's information-sharing structure. The proposed new model has a better fit and provides more accurate provincial-level forecasts with the intervals better reflecting the provincial and regional uncertainty than the Li–Lee model $M_{LL}$. The results of the sensitivity analyses show that the forecasts are relatively robust when changing the grouping assumptions, and confirm that three principal components perform better than two. Overall, the proposed model provides good estimates and reasonable forecasts, and we recommend using the models in both national and subnational mortality modelling.

Based on the mortality forecast, we computed national and subnational life expectancy for China and the United States. The model predicts that the two countries will have the same level of national life expectancy at age 60 in 2030, and that the heterogeneity in subnational life expectancy in the two countries will also be of similar magnitude. However, China has larger forecast intervals for life expectancy at age 60 due to limited data points.

The model we have developed is based on principal components, considering the parsimony and flexibility of this approach. The Bayesian framework can incorporate other functional forms, for example, the Cairns–Blake–Dowd model (Cairns et al. 2006) or models that include cohort effects. Future work may also consider other estimation methods, such

as the Kalman filter, to improve the model fit. Finally, it would also be of interest to study regional annuity pricing and pension liabilities based on the proposed model.

**Author Contributions:** Conceptualization, Q.L., K.H. and X.W.; methodology, Q.L.; software, Q.L.; validation, Q.L., K.H. and X.W.; formal analysis, Q.L.; investigation, Q.L.; resources, Q.L.; data curation, Q.L.; writing—original draft preparation, Q.L.; writing—review and editing, Q.L., K.H. and X.W.; visualization, Q.L.; supervision, K.H. and X.W.; project administration, K.H.; funding acquisition, K.H. and X.W. All authors have read and agreed to the published version of the manuscript.

**Funding:** This research received funding from the Ministry of Education of China (No. 20JZD023) and the National Social Science Fund of China (No. 20AZD075).

**Data Availability Statement:** The data presented in this study are available on request from the corresponding author. The data are not publicly available because part of the data are reserved in archives.

**Acknowledgments:** The authors acknowledge financial support from the Australian Research Council Centre of Excellence in Population Ageing Research (CEPAR) and the China Scholarship Council. This research includes computations using the computational cluster Katana, supported by Research Technology Services at UNSW Sydney. We are also grateful for the comments received from Andrés Villegas and Cheng Wan.

**Conflicts of Interest:** The authors declare no conflict of interest.

## Notes

1   The NBS sums all provincial data to obtain the national data for China. The 1982 census materials include population and deaths by age and gender for China and every province except Hainan (which had not been established), Chongqing (also not established), and Tibet (only population data). The 1990 census materials include population and deaths by age and gender for China and every province except Chongqing province (not established then).

2   We do not focus on the effect of death underreporting in this study but refer interested readers to Lu et al. (2020) for more details of the influence of death underreporting.

3   We use the raw data from the census and sample survey materials to retain the characteristics and information in these data.

4   The 1982 census data were collected in one-year age groups (0, 1, 2, 3, . . . ), with the highest ages ranging from 105 to 140 years old for different provinces. After 1990, census data were collected in one- and five-year age groups (0–4, 5–9, 10–14, . . . ), with the highest age being 100+ years old, except in 1990, when the highest age was 90+ years old.

5   The estimation results are similar for all provinces within a region. Beijing and Gansu provinces represent low mortality provinces and high mortality provinces, respectively. Both provinces have moderate life expectancies within their region (Zhou et al. 2016).

6   The provincial fitting and forecast intervals are similar for different age groups over time. Here, we consider the 0 and 1–4 age groups as representative of early childhood, the 20–24 and 40–44 age groups as representative of middle age, and the 60–64 and 80–84 age groups as representative of old age.

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
