# Peer review of "Subnational Mortality Modelling: A Bayesian Hierarchical Model with Common Factors"

_risks, doi:10.3390/risks9110203_

Round 1
Reviewer 1 Report
My main concern is the data span. It seems that the each subsequent census was carried out in 10 years period so there should be one in 2020. Why wasn't it included? Globalisation of economy makes all the changes occur faster therefore in 10 years a lot of possible differences/changes (medicine develops, pandemics occur...) could happen. This leads to the question whether the conducted research is really reliable.
It is common knowledge that different societies follow different patterns (Chinese/American). I do not fully agree with the utilisation of the model in these two compleletey different societies. Alos, China as a benchmark seems a bit of a stretch.
"There is limited research developing stochastic mortality models for China" - how about other countries? What is the current state-of-the-art in that field?
Author Response
We are very grateful for the referee report we received and the opportunity to revise our manuscript.
Please see the attachment.
Kind regards,
The authors

Reviewer 2 Report
The Bayesian hierarchical model gives a better estimation and forecasting of national and subnational mortality. However, this idea is based on the prior information, which should be controlled. Furthermore, the assumptions of Poisson and Normal distributions are not completely justified. Nevertheless, it provides an alternative method in mortality modelling. The application on China and USA does not bring some additional insight.
Author Response
Please see the attachement.

Reviewer 3 Report
A Bayesian hierarchical framework model to project mortality is proposed. Pool information are used, by using provincial-level mortality data for China. Estimates and forecasts are attained. A comparison life expectancies for China and the United States is provided.
Introduction proposes a complete sketch about the involved dynamics and the relative state of the art.
The study proposes a complete analysis in terms of living place and age about the populations taken into account.
Major remarks:
- Authors have to better define the prime principles reasons about choosing a Bayesian approach.
- The log-scale model, must be derived from data, here the authors, in the initial section of the work, never make a reference to the real data, in order to derive the model taken into account.
- Authors have to specify the reason about considering a three components decomposition, the dimension usually is derived by taking into account data, here the authors start from a theoretical approach.
- The random walk approach, uses results, along its deriving, never considering the data, please specify.
- Subsection 4.1: the authors identify the components respect the real data, I can't understand the methods about the choice of the components.
Minor remarks:
- The authors could use an approach with a confidence level higher than 95%, this could made the result more robust.
Round 2
Reviewer 1 Report
Dear Authors,
thank you for the reply. I still hold the reservation about the time horizon but understand that without the available data you cannot update the research any further.
Reviewer 2 Report
I am satisfied with the revision.